# The Role of the Cerebellum in Advanced Cognitive Processes in Children

**DOI:** 10.3390/biomedicines12081707

**Published:** 2024-08-01

**Authors:** Stefano Mastrangelo, Laura Peruzzi, Antonella Guido, Laura Iuvone, Giorgio Attinà, Alberto Romano, Palma Maurizi, Daniela Pia Rosaria Chieffo, Antonio Ruggiero

**Affiliations:** 1Pediatric Oncology Unit, Fondazione Policlinico Universitario A. Gemelli IRCCS, 00168 Rome, Italy; stefano.mastrangelo@unicatt.it (S.M.); laura.peruzzi@guest.policlinicogemelli.it (L.P.); antonella.guido@guest.policlinicogemelli.it (A.G.); giorgio.attina@policlinicogemelli.it (G.A.); alberto.romano@guest.policlinicogemelli.it (A.R.); palma.maurizi@unicatt.it (P.M.); 2Department of Woman and Child Health and Public Health, Università Cattolica del Sacro Cuore, 00168 Rome, Italy; 3Clinical Psychology Unit, Fondazione Policlinico Universitario A. Gemelli IRCCS, 00168 Rome, Italy; danielapiarosaria.chieffo@policlinicogemelli.it; 4IRCCS Don Gnocchi Foundation, 50143 Florence, Italy; liuvone@dongnocchi.it; 5Department of Life Sciences and Public Health, Università Cattolica del Sacro Cuore, 00168 Rome, Italy

**Keywords:** cerebellar lesions, higher order functions, cognitive disorders, development, procedural learning, children

## Abstract

Over the last several years, a growing body of evidence from anatomical, physiological, and functional neuroimaging studies has increasingly indicated that the cerebellum is actively involved in managing higher order cognitive functions and regulating emotional responses. It has become clear that when children experience congenital or acquired cerebellar lesions, these injuries can lead to a variety of cognitive and emotional disorders, manifesting in different combinations. This underscores the cerebellum’s essential role not only throughout developmental stages but particularly in facilitating learning processes, highlighting its critical importance beyond its traditional association with motor control. Furthermore, the intricate neural circuits within the cerebellum are believed to contribute to the fine-tuning of motor actions and coordination but are also increasingly recognized for their involvement in cognitive processes such as attention, language, and problem solving. Recent research has highlighted the importance of cerebellar health and integrity for optimal functioning across various domains of the human experience.

## 1. Introduction

It has been widely recognized that the cerebellum plays a crucial role in motor control, including the coordination of movements and the acquisition of complex motor skills [1]. However, in recent years, a growing body of evidence has indicated that the cerebellum’s functions extend beyond motor control, also encompassing aspects of cognitive and social functioning [2]. Disruptions to the cerebellum can lead to impairments in a variety of areas such as language, procedural memory, executive functions, visuospatial abilities, and emotional and social behaviors. Consequently, the cerebellum is now acknowledged as a significant player in a range of disorders affecting the central nervous system (CNS), from those involving motor impairment to cognitive and affective disorders [1].

This multifaceted involvement of the cerebellum highlights its importance beyond traditional motor coordination roles, suggesting a more complex integrative function within the brain’s network.

Recent research increasingly recognizes the cerebellum’s integral role in non-motor functions, illustrating that it is actively involved in complex cognitive processes. This is supported by anatomical evidence showing the cerebellum’s extensive neural pathways linking it to various cortical areas involved in higher brain functions. In animal studies, the cerebellum is shown to receive inputs from an array of brain regions associated with cognition and emotion, such as the hypothalamus, parahippocampal gyrus, cingulate gyrus, superior temporal cortex, posterior parietal cortex, and the prefrontal cortex; in humans, these findings have been corroborated through advanced MRI and tractography [1,3]. These data have also been verified through a study using fMRI [4]. Moreover, the construction of these functional networks occurs gradually over time, beginning with the integration of sensory motor functions and later expanding to include associative regions, underscoring the cerebellum’s evolving involvement in cognitive tasks [3].

This extended development timeline suggests that the cerebellum plays a critical role in the maturation of cognitive functions well into adolescence. These connections within the brain are typically finalized by the age of 9 years, yet there is evidence suggesting that they start to function to some extent even before that, as shown by the early presentation of neurocognitive deficits in children suffering from congenital or acquired cerebellar damage [3]. The intricate relationship between the cerebellum and higher brain structures is further underscored by their concurrent evolutionary progression and notably prolonged development phase, making them the slowest growing areas within the brain [5,6]. This extended period of maturation increases the cerebellum’s vulnerability to various pathologies that can arise early in development, compared with other brain structures that mature more rapidly. Consequently, this heightened susceptibility could shed light on the cerebellum’s frequent involvement in a wide array of pediatric neurological conditions, particularly noting the propensity for the vermis to exhibit hypoplastic features more so than the cerebellar hemispheres, with specific vermian lobes being affected while others remain unaffected.

Numerous studies on healthy adults, as well as those with congenital and acquired cerebellar lesions, have solidified the understanding that the cerebellum plays a crucial role in orchestrating higher brain functions beyond its traditional association with motor control [7,8]. This evidence has progressively broadened the perspective on the cerebellum’s involvement, also emphasizing its significance in the organization of non-motor functions in both adults and children. Interestingly, recent research has not only confirmed the cerebellum’s participation in non-motor language functions but has also specifically shown that verbal fluency is adversely affected by lesions in the deep cerebellar nuclei, regardless of any motor speech impairments [9,10,11]. This indicates that the cerebellum plays a crucial role not only in coordinating physical movements but also in facilitating aspects of cognitive processing, such as language.

While cerebellar malformations and dysfunctions are commonly observed in congenital diseases and various developmental disorders during childhood, instances of acquired cerebellar lesions in children remain relatively rare [12]. Studying acquired cerebellar lesions enables us to broaden our understanding of the cerebellum’s impact on cognitive and emotional/affective development. 

Moreover, it allows us to explore the extent of its functional reorganization following a lesion. Therefore, the importance of the cerebellum in cognition and emotion is now acknowledged in neurological, psychiatric, and psychological research. Additionally, its impact on cognitive, emotional, and behavioral functions creates new opportunities for comprehending and addressing various neurological and psychiatric disorders in both adults and children.

## 2. Congenital Cerebellar Lesions 

Cerebellar malformations occur in an array of over 200 neurogenetic syndromes, such as Williams syndrome and Fragile X syndrome, and affect several metabolic and degenerative diseases including neuroaxonal dystrophy and ceroidolipofuscinosis. These congenital malformations can lead to profound neuropsychological challenges, which often manifest as intellectual disabilities, significant language disorders, and behavioral changes that can be as severe as an autistic phenotype [3,13]. In conditions such as autism spectrum disorders, attention deficit hyperactivity disorder, and developmental dyslexia, cerebellar dysfunction is a common finding, with issues in structural and functional connectivity like hypoplasia and decreased cerebellar volume being correlated with symptom severity and core behaviors of these disorders [14,15]. Particularly, the reduction in grey matter in certain cerebellar areas has been consistently observed in populations with dyslexia, underlining the crucial role of the cerebellum in cognitive functions and neurodevelopmental conditions [12]. Therefore, early identification and intervention focusing on the effects of cerebellar abnormalities on cognitive, emotional, and behavioral growth and performance could potentially improve some of the cognitive, emotional, and behavioral challenges linked to these disorders. As a result, early detection and precise treatment could enhance the well-being of both the patient and their caregivers.

## 3. Acquired Cerebellar Lesions 

When adults develop lesions in the cerebellum, they may exhibit a complex set of behavioral and cognitive impairments known as “cerebellar cognitive affective syndrome (CCAS)”. This syndrome involves difficulties in fundamental areas such as executive functions, visuospatial cognition, language, and personality (emotion/behavior). Lesions in the posterior lobe of the cerebellum result in challenges with executive functioning, such as distractibility or lack of attention, deficits in working memory, planning, problem solving, perseveration, response inhibition, task switching, acquiring new knowledge, and using metacognitive strategies. This leads to difficulties in complex visuospatial processes like mental rotation and visuospatial organization, and language issues, including dysprosodia, agrammatism, syntax impairment, verbal fluency deficits, and slight difficulty in word finding [8]. Changes in personality, either emotional blunting or displaying disinhibited and inappropriate behavior, are associated with lesions in the vermal and paravermal regions.

Neurobehavioral diagnostic profiles have been categorized as disorders of attentional and emotional control, autism, and psychosis spectrum disorder, according to a study by [16]. In the year 2000, CCAS was identified in children, showing similar patterns of impairment as seen in children with cerebellar lesions, especially after tumor treatment or surgery [17,18]. Changes in cognitive function and behavior have been documented in children following surgical treatment for cerebellar tumors in the hemispheres or vermis. These patients exhibit problems with executive function, visuospatial and language skills, social and communication abilities, as well as mood alterations. This clinical image may also manifest with or without post-operative mutism syndrome, which is a potential complication seen mainly in children with a prevalence ranging from 8% to 31% after surgical treatment for posterior fossa brain tumors, such as medulloblastoma [19,20,21]. Approximately 25% of cases exhibit mutism, along with motor speech deficits and other neurological symptoms like difficulty swallowing and lack of coordination, as well as emotional and behavioral changes. The syndrome is believed to be caused by damage to the cerebellum, which is responsible for motor coordination, balance, behavior, and speech.

Levisohn and colleagues, who studied 19 children following surgical treatment for cerebellar tumors, identified comparable deficits affecting executive functions, verbal memory, visuospatial and expressive language skills, and the ability to regulate emotions in communication, which underscores the similarities between the effects of cerebellar lesions in both adults and children (Figure 1) [17].

Furthermore, the research highlighted the connection between damage to the cerebellar vermis and a range of behavioral and cognitive impairments, including issues with affect regulation, cognitive functions, and language abilities—especially when the damage was extensive. Further research on 26 children who had surgery for tumors in the cerebellar hemispheres or vermis confirmed two different outcomes from vermal lesions: one causing post-surgical mutism followed by speech issues, and another leading to social and communication problems, ranging from irritability to autism-like symptoms [18]. This supports Schmahmann’s theory that the vermis plays a key role in the cerebellar limbic system, which regulates emotional and behavioral functions. Taddei and coworkers recently discussed a case report describing a pediatric patient who had surgery to remove the postero-inferior lobules of the vermal due to medulloblastoma. Researchers documented severe post-surgery cerebellar cognitive affective syndrome (CCAS), where a 9-year-old girl exhibited cognitive and executive function impairments and behavioral changes. Social abilities and communication skills were also affected, and the clinic profile resembled that of individuals with autism spectrum disorder [22]. The long-term social impairment was documented after three years of follow-up, while cognitive and neuropsychological functioning (including language and executive functions) showed improvement over time. This case report detailed a prolonged decrease in social functioning. The vermal area’s role in affectivity, social cognition, and emotions is backed by its connections with the vermal area and the fatigued nucleus with cortical, subcortical, limbic, associative, and paralimbic regions [23], as shown in studies on acquired cerebellar lesions [24]. Additionally, it has been found that the surgical removal of tumors from the right cerebellar hemisphere often results in verbal intelligence, auditory sequential memory, and language deficits, whereas those from the left hemisphere largely affected non-verbal skills and notably impaired executive functions, suggesting that neurosurgical procedures in these regions can lead to side-specific neuropsychological deficits [18,19,20,21,22,23,24,25,26]. Starowicz-Filip et al. analyzed the connection between the side of cerebellar damage and the level of deficits in complex visuospatial processes [27], while Di Rocco and colleagues highlighted that a pediatric patient with tumors in the left cerebellar hemisphere exhibited deficits in preoperative visual spatial skills [28]. Moreover, the study highlighted that patients with left hemisphere diseases often exhibit more frequent occurrences of selective language planning defects or language disabilities. This is particularly noticeable in cases where there is an initial bilateral representation of language, as observed in healthy children and individuals with congenital or early unilateral cerebral lesions. In fact, research has shown that language processing relies on an interconnected circuit involving the left supratentorial language areas and the right posterior lateral cerebellar hemisphere, specifically lobule VI, lobule VII, Crus I, and Crus II. These findings are supported by another study that showed reduced activity in the lateralization function of the cerebro-cerebellar language system in the presence of a right hemispheric lesion. The damage in the postero-lateral cerebellum appeared to be related to less efficient language performance, serving as an indicator of the system’s functionality [29]. Notably, this study also, for the first time, investigated the functional reorganization of language after an early right lesion in the cerebellum. Using an fMRI paradigm of a phonological task among 6 children who had undergone surgery for right cerebellar astrocytoma and 15 typically developing children, the study found right cerebellar and left frontal activations in healthy controls, with a high variability in reorganizational patterns among patients with early right cerebellar damage. This high variability in reorganization could be attributed to factors such as the extent and location of the lesion and the timing of surgical therapy, as proposed by Riva and colleagues [29]. Our previous research, focusing on 83 children with supratentorial hemispheric, supratentorial midline, and cerebellar tumors evaluated prior to any treatment or surgery, corroborates these findings, revealing a notable impairment in phonological working memory among patients with right-side cerebellar tumors, whereas left-sided tumors tended to impact visual spatial skills [30]. These data could indicate a diaschisis relationship between the right cerebellar hemisphere and the left frontal cortex. Put differently, damage to the cerebellum would result in a decrease in functioning of the brain areas responsible for language processing due to the “crossed cerebello-cortical diaschisis” of the dentate-thalamic-cortical pathway [25]. “Temporary disruption of connections between the cerebellum and contralateral frontal areas, known as cross-cerebral diaschisis, may be the neuropathological mechanism behind language deficits.” Based on this, it is suggested that the cerebellum does not have a direct role in language production but instead serves as a regulator for ensuring its proper implementation. The dysfunction of the cerebello-cerebro-cortical circuits caused by a lesion in the right cerebellar hemisphere can account for the poor performance in verbal memory tasks. Gottwald and colleagues suggested that the right brain hemisphere is involved in verbal working memory, while the left cerebellar hemisphere is important for non-verbal or visuospatial working memory [31]. Multiple studies using functional neuroimaging highlight the significance of the left hemisphere in visuospatial working memory [32], with Hoang et al. also confirming this discovery [33,34]. The task of verbal memory involves the need for phonological recoding, suggesting a possible impairment in the articulatory rehearsal process. This method would enable the repetition of the stored information to solidify it and aid in its immediate recollection [35]. According to research, it has been established that the process of rehearsal in both adults [36] and children [13] activates particular circuits in the right hemisphere of the cerebellum [36].

Additionally, studies have discovered that numerous children with cerebellar tumors also exhibit slow information processing speeds [37]. Hogan et al. [33] found a decreased index processing speed score on the intelligence scale in their research study. In their study, Riva and Giorgi [38] discovered that children with cerebellar tumors exhibited significant delays in completing tasks that required timing. This was described as the result of the close proximity of the cerebellar lesion to the “ascending reticular activating system” found in the brainstem, which is responsible for regulating the waking state and various attentional functions [39]. Riva and his team [39] also linked the slow behavior of the children in the study to a lack of proper function of the intrinsic cerebellar structure, consisting of micro modular systems, where each microprocessor-like module operates simultaneously with the others; alignment in parallel would result in a network with a high processing speed. The potential to add modules in quick succession could lead to the circuit achieving high levels of speed in execution. Therefore, if there is damage to any part of the network, it will affect the speed at which tasks are performed [40]. In the event of a large amount of damage, not only specific tasks will be compromised [41]. The impairment in executive functions can be linked to damage in the cerebellum, which regulates the processing of motor functions and higher brain functions by interacting with cerebral cortical areas through feedforward and feedback circuits.

Additionally, individuals with damage to the cerebellum may experience challenges in procedural learning as well. Molinari and his team [42] discovered that adult patients showed impaired sequential implicit learning. This condition can also occur in young patients. In a study conducted by Quintero-Gallego et al., involving 18 children diagnosed with astrocytoma and medulloblastoma cerebellar tumors, it was discovered that these children exhibited significant difficulties in procedural learning [43]. 

These patients need explicit and specific strategies to achieve this task. This means selective and specific cerebellar commitment to the formation of procedural strategies. It is hypothesized that the basic mechanism of cerebellar intervention is to analyze the sequence relationship between different events [44]. According to this hypothesis, Molinari et al. [10] argue that the specific role of the cerebellar in procedural processing is to recognize the existence of sequences, i.e., the cerebellar would detect by “comparison behavior” between new information inputs of the fronto-parietal-temporal regions and those expected and predicted by the cerebellar and temporarily in memory, detecting and signaling errors to the cerebral cortex. This function would also consider the findings of other fMRI studies, in which, during procedural learning, the activation of the cerebellar seems to increase significantly during the first phase of the task, i.e., information encryption, rather than their automatic recall. Consequently, a brain that should interact in the first stages of learning each function, such as walking and language, functions that were first learned completely unconscious and then become intentional. It is precisely the first phase that allows the formation of implicit knowledge of events [45]. Changes in the mental representation of reality by the brain only begin without being conscious when the cortex regions (fronto-temporo-parietal) receive communications of such changes from the brain circuits [46,47].

Any change in the “mental model” is reflected in changes in the thought process concerning facts, people, things, and places with which the subject is in contact every day. Thus, thinking activities are composed of implicit and explicit processing processes—the first takes place in the brain and the second in the brain cortex. Given the impairments in executive functions, visual-spatial abilities (visuospatial cognition), and linguistic abilities, as well as the potential for decreased information processing speed and emotional/behavioral challenges that children may exhibit following an acquired cerebellar lesion, early and targeted intervention can greatly enhance the prospects and outcomes of children who survive cerebellar tumors and experience chronic side effects [48,49]. 

## 4. Discussion

Previously, the cerebellum was thought to mainly play a role in sensorimotor coordination and performance. Recent research shows that the cerebellum is essential for various higher brain functions, as evidenced by clinical, neuroanatomical, and neuroimaging studies linking it to cognitive, affective, social, and behavioral processes. Research using functional neuroimaging has identified a separated network of brain regions connecting the cerebellum with the supratentorial, paralimbic, and association motor cortex. This network is believed to underlie the cognitive and emotional functions of the cerebellum [50]. The cerebellar functions are determined by the anatomical connections of different parts of the cerebellar cortex with various cortical areas related to higher brain functions. Therefore, certain cognitive deficits and motor/affective issues can be directly linked to specific lesions within the cerebellum, highlighting the importance of understanding its functional topography within the brain [51,52,53]. Consequently, issues in these connections within the cerebellum have been linked to different neuropsychiatric disorders, underscoring the cerebellum’s significance in both normal brain function and adult and pediatric diseases. An understanding of the cerebellar functional topography is essential in the study of patients with acquired cerebellar lesions, particularly in pediatric patients who experience deficits at an early stage [53,54]. The presence of early acquired lesions provides an opportunity to investigate the capacity of the cerebellum to reorganize its functions following damage. However, in the case of language processing, it remains unclear and unknown how the reorganization occurs after an early right cerebellar lesion [29]. Some theories suggest that there is initially a bilateral contribution from both cerebellar hemispheres to the language system, followed by a specialization towards the right cerebellum, which aligns with the left cerebral lateralization [55]. 

Research in the field of pediatrics, although not abundant, underscores the cerebellum’s crucial influence on cognitive and emotional development in the early years. Specifically, the study of pediatric patients with acquired cerebellar lesions provides valuable insights into the cerebellum’s function and role, as these lesions are typically localized, offering clear differentiation from more generalized brain conditions. Comparative studies highlight that individuals with congenital cerebellar lesions often face more significant cognitive, behavioral, and motor challenges, including ataxia and coordination issues, than those with acquired lesions who tend to exhibit milder impairments [42,43]. Damage to the cerebellum in the early stages could adversely affect the circuits that connect with the supratentorial areas, leading to changes in structure and function. These areas are responsible for higher cognitive functions. Notably, research involving both children and adults with cerebellar damage has shown that focal lesions on the cerebellum adversely affect procedural learning, suggesting the cerebellum plays a pivotal role in supporting implicit or procedural learning processes [42,43], as confirmed by Bianco et al. [56]. Recent studies highlight the cerebellum’s key role in detecting and recognizing patterns in sequential events, which is fundamental to the process of making new cognitive procedures automatic, an aspect of implicit learning [43,44,45,46,47,48,49,50,51,52,53,54,55,56,57]. Such learning is not only essential for mastering motor skills but is also critical in the development of cognitive and linguistic abilities, including phonological processing and literacy, facilitating effortless reading and writing. These insights help to explain the difficulties experienced by people with conditions such as dyslexia, which can be associated with cerebellar impairments, especially in its lateral regions and connected networks. Additionally, while the cerebellum’s involvement in learning new tasks is crucial, this role wanes as the tasks become more automatic and are taken over by specialized cortical networks, underscoring the cerebellum’s vital influence in the early stages of learning and overall developmental growth [12,42,58]. 

In summary, the cerebellum plays a crucial role in the development of motor skills and cognitive functions, regulating emotional responses and social-behavioral capacities. This importance is highlighted by the observation that children with cerebellar lesions acquired later in life tend to exhibit milder cognitive and motor deficits compared with those with congenital cerebellar lesions, who often face a more complex spectrum of challenges, including intellectual disabilities, language problems, behavioral issues, and motor disorders. 

Additionally, in adults, cerebellar lesions typically result in transient cognitive deficits, whereas, in very young children, such injuries may halt progress in developing skills at a rate comparable to their peers. This suggests that the cerebellum’s capacity for functional reorganization differs significantly from that of the cerebral cortex, particularly in younger individuals [29,59].

## 5. Conclusions

In recent years, there has been an increasing amount of evidence suggesting that the functions of the cerebellum go beyond motor control. It has been found that the cerebellum also plays a role in cognitive, emotional, and social functioning. This is supported by both anatomical and functional evidence, which show that the cerebellum is connected to various cortical regions involved in higher brain functions. Furthermore, research has shown that the development of these functional networks in the cerebellum happens gradually over time. It starts with the integration of sensory-motor functions and later extends to include associative areas. This highlights the evolving role of the cerebellum in cognitive processes and tests. This extended development timeline suggests that the cerebellum plays a critical role in the maturation of cognitive functions throughout adolescence. The prolonged maturation period increases the vulnerability of the cerebellum to various disorders that can emerge early in development, unlike other brain structures that mature more quickly. This heightened susceptibility could provide insight into the cerebellum’s frequent involvement in a wide range of pediatric neurological conditions.

Data on clinical studies have shown that disruptions to the cerebellar regions can result in impairments across various domains including executive functions, procedural memory, visuospatial abilities, language, and emotional and social behaviors. Studies on pediatric patients with cerebellar damage highlight the significant impact of the cerebellum on cognitive and emotional development during early childhood, emphasizing its central and essential role in the development of motor skills, cognitive functions, emotional regulation, and social-behavioral abilities. 

In summary, the cerebellum plays a critical role in both cognitive and affective development, particularly during the early stages of life, when it supports cognitive and motor learning processes. Studying the long-term cognitive outcomes of older versus younger children could shed light on a potential “critical period” during which the cerebellum’s development is paramount for mastering complex functions. Understanding the age at which specific competencies are acquired may serve as a marker for the cutoff point of cerebellar involvement in these developmental processes. 

Therefore, future research that includes comparative studies of children who have suffered cerebellar lesions at various ages could offer invaluable insights into the cerebellum’s significant contributions to developmental milestones. Furthermore, studying acquired cerebellar lesions allows us to explore and understand the extent of functional reorganization following a lesion of the cerebellum, which until now has been unclear. 

In recent years, the remarkable progress in neuroscience has had a profound impact on our comprehension of the impact of cerebellar structure on human behavior. Undoubtedly, this transformation will bring about substantial consequences for rehabilitation. Consequently, conducting research on the profiles of adult and pediatric patients with cerebellar lesions can yield valuable insights into rehabilitation practices. This research can enable us to provide our patients with new and innovative treatment options that will enhance their quality of life from a holistic perspective, encompassing the biological, psychological, and social aspects of their well-being.

## Figures and Tables

**Figure 1 biomedicines-12-01707-f001:**
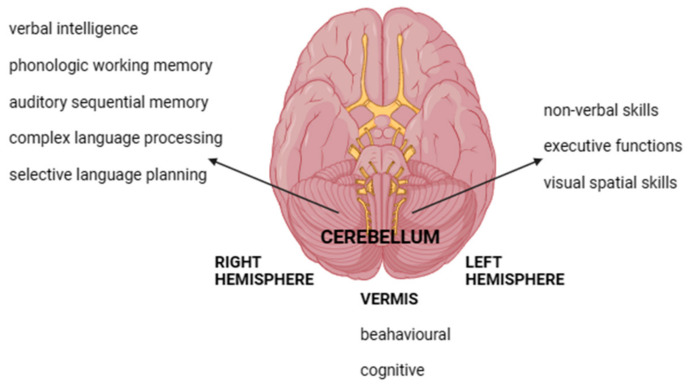
Specific cerebellar functions impaired by acquired lesions to different sites.

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
