# Peer review of "The Role of the Cerebellum in Advanced Cognitive Processes in Children"

_biomedicines, 2024, doi:10.3390/biomedicines12081707_

Round 1

Reviewer 1 Report

Comments and Suggestions for Authors

Review paper entitled "The Role of the Cerebellum in Advanced Cognitive Processes in Children" by Stefano Mastrangelo and colleagues is interesting topic but lack of several component from beginning to the end of the paper. 

There are plenty of papers are available like 

https://linkinghub.elsevier.com/retrieve/pii/S1052-5149(16)30001-6https://neuro.psychiatryonline.org/doi/full/10.1176/jnp.12.2.193

https://link.springer.com/article/10.1007/s00415-021-10486-w

Secondly, paper is designed not having any flow and very fewer studies are included in the review article and is not providing any new information. 

Recent papers are very less and warrants extensive literature survey.

Discussion on mechanistic approach is missing. 

Comments on the Quality of English Language

No major issue

Author Response

Reviewer 1
Review paper entitled "The Role of the Cerebellum in Advanced Cognitive Processes in Children" by Stefano Mastrangelo and colleagues is interesting topic but lack of several component from beginning to the end of the paper.

There are plenty of papers are available like

https://linkinghub.elsevier.com/retrieve/pii/S1052-5149(16)30001-6https://neuro.psychiatryonline.org/doi/full/10.1176/jnp.12.2.193

https://link.springer.com/article/10.1007/s00415-021-10486-w

Secondly, paper is designed not having any flow and very fewer studies are included in the review article and is not providing any new information.

Recent papers are very less and warrants extensive literature survey.

Discussion on mechanistic approach is missing.

R. Thank you very much for the comment. The manuscript has been completely revised: some sentences have been eliminated in an attempt to structure a flow of information and include further new details on the role of the cerebellum. The Discussion and Conclusions section have been extensively revised.

Reviewer 2 Report

Comments and Suggestions for Authors

The paper states to present the role of the cerebellum in advanced cognitive processes in children. This is true for chapters 1 and 2 and most parts of chapter 3. At the end of chapter 3, the papers changes its topic and deals with cancer surgery and therapy: Discussion is on the advance of cancer surgery and therapy of pediatric patients. At line 206 it returns to the topic of the paper (the cerebellum's function and role). 

Yes, research on the cerebellum's function and role is closely related to tumor surgery and therapy. However, the paper is better focusing on one theme, e.g. the cerebellum's function and role, with surgery and therapy presented less dominant. In the current version, the broad discussion of surgery and therapy (plus its advances etc.) shows up late and has a disturbing effect. 

Author Response

Reviewer 2
The paper states to present the role of the cerebellum in advanced cognitive processes in children. This is true for chapters 1 and 2 and most parts of chapter 3. At the end of chapter 3, the papers changes its topic and deals with cancer surgery and therapy: Discussion is on the advance of cancer surgery and therapy of pediatric patients. At line 206 it returns to the topic of the paper (the cerebellum's function and role).

Yes, research on the cerebellum's function and role is closely related to tumor surgery and therapy. However, the paper is better focusing on one theme, e.g. the cerebellum's function and role, with surgery and therapy presented less dominant. In the current version, the broad discussion of surgery and therapy (plus its advances etc.) shows up late and has a disturbing effect.

R. Thank you very much for your suggestions. We have completely revised the manuscript by deleting some sentences and adding/rewriting some other parts.

Round 2

Reviewer 1 Report

Comments and Suggestions for Authors

Authors improved paper but there are still some concern.

1. Has your designed figure 1 by yourself or adopted from somewhere else? Need to mention, also make sure that there is no copyright issue.

2. Conclusion is too long, need to be specific and concise.

3. There are still typo and grammatical errors that need to be corrected. 

Comments on the Quality of English Language

none

Author Response

REVIEWER 1

Authors improved paper but there are still some concern.

  1. Has your designed figure 1 by yourself or adopted from somewhere else? Need to mention, also make sure that there is no copyright issue.

        R. The figure 1 is not subject to copyright as it was designed by us

  1. Conclusion is too long, need to be specific and concise.

        R. The Conclusion section has been reviewed and shortened

  1. There are still typo and grammatical errors that need to be corrected.

         R. Typo and grammatical errors have been corrected